# Non-ROS-Mediated Cytotoxicity of ZnO and CuO in ML-1 and CA77 Thyroid Cancer Cell Lines

**DOI:** 10.3390/ijms24044055

**Published:** 2023-02-17

**Authors:** Alyse N. Peters, Nakaja A. Weaver, Kathryn S. Monahan, Kyoungtae Kim

**Affiliations:** 1Department of Biology, Missouri State University, 901 S National, Springfield, MO 65897, USA; 2Department of Chemistry, Missouri State University, Springfield, MO 65897, USA

**Keywords:** MONPs, ZnO, CuO, ML-1 cells, CA77 cells, toxicity, apoptosis, RNAseq

## Abstract

Metal oxide nanoparticles (MONPs) are widely used in agriculture and food development but there is little understanding of how MONPs, including ZnO, CuO, TiO_2_, and SnO_2_, impact human health and the environment. Our growth assay revealed that none of these (up to 100 µg/mL) negatively affect viability in the budding yeast, *Saccharomyces cerevisiae*. In contrast, both human thyroid cancer cells (ML-1) and rat medullary thyroid cancer cells (CA77) displayed a significant reduction in cell viability with the treatment of CuO and ZnO. The production of reactive oxygen species (ROS) in these cell lines, when treated with CuO and ZnO, was found to be not significantly altered. However, levels of apoptosis with ZnO and CuO were increased, which led us to conclude that the decreased cell viability is mainly caused by non-ROS-mediated cell death. Consistently, data from our RNAseq studies identified differentially regulated pathways associated with inflammation, Wnt, and cadherin signaling across both cell lines, ML-1, and CA77, after ZnO or CuO MONP treatment. Results from gene studies further support non-ROS-mediated apoptosis being the main factor behind decreased cell viability. Together, these findings provide unique evidence that the apoptosis in response to treatment of CuO and ZnO in these thyroid cancer cells was not mainly due to oxidative stress, but to the alteration of a range of signal cascades that promotes cell death.

## 1. Introduction

Metals such as gold, silver, titanium, copper, zinc, and aluminum have all been used to synthesize nanomaterials, as well as their oxides, i.e., silver oxide, copper oxide, magnesium oxide, calcium oxide, and zinc oxide. These common nanomaterials exhibit magnetic, fluorescence quenching, and dielectric properties [1,2]. In particular, MONPs are no more than 100 nm in size and are one of the most studied nanomaterials against multidrug-resistant bacteria [3,4]. Importantly, the small size of MONPs increases the surface/volume ratio, which improves their stability and the ability to interact with different cellular components [5]. Furthermore, MONPs are very popular because of their lower price and simple synthesis process [6]. 

Many researchers have uncovered the antibacterial properties of MONPs, which can make an impact on bacterial growth and reproduction [7]. Infectious bacterial cells accumulate on hospital bed sheets, patient gowns, and uniforms, and can be one of the main causes of hospital-acquired infections (HAIs). To combat HAIs, CuO, ZnO, and TiO_2_ MONPs of different sizes were all tested on *S. aureus*, *B. subtilis*, *P. aeruginosa*, and *E. coli* [7]. Among the tested MONPs, ZnO and CuO MONPs caused the greatest decrease in cell viability in bacterial strains. In particular, CuO MONPs of 10–14 nm caused decreased viability in *S. aureus* and *B. subtilis*, while ZnO NPs of 12–15.5 nm caused decreased viability in *P. aeruginosa* and *E. coli* [7]. Overall, all the MONPs tested clearly had the ability to counteract the proliferation of bacterial strains. Frenk et al. assessed CuO and Fe_2_O_3_ in a study to understand the potential impact of these MONPs on soil-living bacteria and found that CuO has a more severe negative impact on hydrolytic activity and soil oxidative potential [8]. Both CuO and Fe_2_O_3_ used in the study were about the same size (smaller than 50 nm), and therefore, this investigation has limited information on the impact of MONP size on cell toxicity. Zhang et al. evaluate TiO_2_ with several different sizes on human epidermal keratinocytes and found that only TiO_2_ of 27.5 nm induced modest cytotoxicity compared to larger particles [9]. A similar study focused on the shape of MONPs rather than size; they compared rod-shaped and sphere-shaped Fe_2_O_3_ and CuO [10]. From this study, both rod-shaped Fe_2_O_3_ and CuO exhibited more toxic effects on cell viability and ROS production than spherical MONPs [10]. They proposed that the greater surface area of rod-shaped MONPs compared to spherical MONPs is the main cause of the increased toxicity [10].

In vitro cell culture is a useful tool to gain insight into underlying toxicity mechanisms. Many researchers are concerned with MONPs’ (ZnO, CuO, TiO_2_, and SnO_2_, specifically) impact on human health and the environment [11]. A recent study evaluated TiO_2_ on different aquatic cell lines (RTG-2, PLHC-1, RTH-149, and RTL-W1) and tested their cytotoxicity [12]. The mitochondria activity, plasma membrane integrity, and lysosome function in each cell line stated above after TiO_2_ treatment (100 µg/mL and lower) were detected to be normal [12], indicating low cytotoxicity of TiO_2_ MONPs. Similarly, SnO_2_ was reported to have negligible cytotoxic effects [13]. Ahamed et al. investigated SnO_2_, concentrations ranging from 10–200 µg/mL, on breast cancer MCF-7 cells [13]. Interestingly, SnO_2_ alone conveyed no toxic effects but when paired with ZnO nanocomposite (SnO_2_-ZnO/rGO NPs) the resulting toxicity was greatly increased [13]. In contrast to the previously reported non-toxic effects of TiO_2_ and SnO_2_, ZnO NPs are one of the most toxic MONPs. In a 2019 study, HaCaT keratinocytes were treated with ZnO MONPs and showed increased toxicity due to the dissolution of Zn^2+^ ions, emphasizing that the cytotoxicity is caused by the ability of ions to internalize by endocytosis [14]. Similarly, Mahmood et al. stated that CuO MONPs elicit significant cytotoxicity in MCF-7 cells by activating the apoptotic signal cascade [15].

Given that CuO and ZnO led to more toxicity over TiO_2_ and SnO_2_ against mammalian cells as stated above, most researchers gave attention to the study of the toxic mechanisms in response to ZnO and CuO MONPs in both in vitro and in vivo experiments. However, these MONPs’ impact on animal and human thyroid cancer cells has not been documented at all. In addition, the literature on MONP’s effects on fungal cells is scarce. Therefore, we evaluated the toxic impact of MONPs such as ZnO, CuO, TiO_2,_ and SnO_2_, against two thyroid cancer cell lines, ML-1 (human thyroid cancer) and CA77 (rat medullary thyroid carcinoma), and the budding yeast *Saccharomyces cerevisiae* to obtain broader understanding to the cytotoxicity of MONPs and their potential environmental implications. 

## 2. Results 

### 2.1. Scanning Transmission Electron Microscopy with ZnO and CuO NPs

As stated in the method section, TiO_2_ and SnO_2_ had no significant impact on cell viability, thus we decided to focus on investigating the physicochemical properties of CuO and ZnO for the current study. STEM images for each nanomaterial show that the CuO and ZnO nanomaterial sizes are in agreement with that of the data sheet values; 40 nm and 35–45 nm respectively (Figure 1).

### 2.2. X-ray Diffraction of ZnO and CuO NPs 

Using Bruker’s Evaluation software, we analyzed X-ray diffraction (XRD) patterns (Figure 2) of the ZnO and CuO nanomaterials that we used in the current study and confirmed that each nanomaterial fitted to the standard crystallographic pattern of the corresponding nanomaterials, suggesting that these nanomaterials show high crystallinity. The nanomaterials utilized here are truly the oxide forms of Zn and Cu; the peaks identified the material based on the Miller index and peak fitting. 

### 2.3. Yeast Viability Was Not Compromised by the Treatment of ZnO, CuO, TiO_2_, and SnO_2_

Absorbance-based growth assays were performed to better understand the potential toxicity of ZnO, CuO, SnO_2_, and TiO_2_ on the biological organism Saccharomyces cerevisiae. After analyzing the optical density of MONP-treated cells at 594 nm over a period of 24 h, we found that all MONPs did not reduce the growth of Saccharomyces cerevisiae at concentrations up to 100 µg/mL (Figure 3A–H). Rather, as shown in Figure 3B,F, ZnO and SnO_2_ caused slightly increased growth when compared with non-treated groups (0 µg/mL). These results are expected because budding yeast has a cell wall which is hypothesized to be a protective layer that keeps MONPs from accessing cell cytoplasm. 

### 2.4. XTT Cell Viability Assay

To investigate which metal nanoparticles, ZnO, CuO, TiO_2_, or SnO_2_, are potentially harmful to ML-1 thyroid cancer cells, an XTT cell viability assay was conducted. ML-1 cells were incubated with concentrations of 4 µg/mL, 20 µg/mL, and 100 µg/mL for each of the four MONPs. After 7 h of incubation with an XTT activation reagent, ML-1 viability correlates to measured absorbance levels. Five groups (100 µg/mL of ZnO, 100 µg/mL of CuO, and all three DMSO groups) showed *p* < 0.001 as compared with non-treated control (NTC). In particular, at the concentration of 100 µg/mL, both ZnO and CuO nanoparticles had significantly decreased cell viability (Figure 4). Based on this XTT assay, ZnO and CuO were chosen as the most toxic nanoparticles out of the four and were used for the following experiments. 

Two MONPs, ZnO and CuO, showed markedly decreased cell viability when treated to ML-1 cells (Figure 4). To determine IC_50_ values, differing amounts (20 µg/mL, 40 µg/mL, 60 µg/mL, 80 µg/mL, and 100 µg/mL) of both ZnO and CuO were treated to ML-1 cells (Figure 5A). After 7 h of incubation with an XTT activation reagent, cells with 40 µg/mL of ZnO displayed a significantly decreased cell viability (*p* < 0.01), while 60 µg/mL of CuO had a comparable level of viability reduction to 40 µg/mL of ZnO-treated cells (Figure 5A, *p* < 0.05). Using the AAT Bioquest website, we determined IC_50_ values of ZnO and CuO for ML-1 cells: 22.8 µg/mL for ZnO and 45.5 µg/mL for CuO.

CA77 rat medullary thyroid carcinoma cells displayed similar levels of viability loss when compared to that of ML-1 human thyroid cancer. A comparison of these two cell lines can give an insight as to which cell line is more sensitive to MONP treatment. An additional XTT cell viability was conducted on CA77 cells with a similar treatment: 20 µg/mL, 40 µg/mL, 60 µg/mL, 80 µg/mL, and 100 µg/mL for both ZnO and CuO (Figure 5B). A concentration of 20 µg/mL for both CuO and ZnO was the lowest concentration in which the viability of CA77 cells was inhibited (Figure 5B, *p* < 0.05). The next higher concentrations of these MONPs also caused decreased cell viability, with statistically significant *p* values (Figure 5B, *p* < 0.01). IC_50_ values were calculated in a manner identical to that with the ML-1 cells: 68.2 µg/mL of ZnO and 72.8 µg/mL of CuO for CA77 cells. 

Incubation of MONPs with mouse fibroblast cells can help determine if they could be used as an anticancer treatment. An important aspect of cancer therapy we should consider before its application to cancer is if they are able to target only cancer cells. If MONPs also lead to a reduction in fibroblast cell viability, then they would be a poor option for anti-cancer treatment. It was found that fibroblast cells showed decreased cell viability with ZnO treatment (Figure 5C, *p* < 0.05). However, CuO NPs showed similar trends compared to NTC. ML-1 cells were determined to be the most sensitive to ZnO and CuO treatment compared to CA77 cells.

### 2.5. Reactive Oxygen Species

Reactive oxygen species (ROS) are free radicals that largely contribute to disturbing the cell in stressful environments. Superoxide level was tested in ML-1 and CA77 cells using dihydroethidium (DHE). Cells were treated for 24 h with IC_50_ values of ZnO and CuO calculated for ML-1 cells; 22.4 µg/mL of ZnO or 45.5 µg/mL of CuO. Samples were incubated with DHE and measured with flow cytometry. Observing the fluorescence intensity of DHE of each cell sample, we determined the percentage of cells with oxidative stress. Three samples for each treatment group were averaged to visualize the differences between them. A solution of 20% DMSO was used as a positive control (Figure 6A–D). In the ML-1 treatment for 24 h with ZnO or CuO, ROS production was not statistically significantly different compared to the non-treated control (Figure 6A). Median fluorescent intensity showed an increase in superoxide production for the positive control (*p* < 0.05) (Figure 6B), but no significant alteration in superoxide production for both treatments (22.4 µg/mL of ZnO and 45.5 µg/mL of CuO) (Figure 6B). ML-1 cells were then treated for 4 h with these MONPs because 24 h of incubation with these MONPs resulted in a cell viability decrease (Figure 4 and Figure 5), which possibly indicates that superoxide levels could be elevated at an earlier time point. However, even a 4 h (Figure 6C) or a 30 min treatment (Figure 6D) with ZnO or CuO resulted in no significant alteration of superoxide levels. CA77 cells were treated with 68.9 µg/mL of ZnO and 72.3 µg/mL of CuO for 24 h (Figure 6E), revealing no significant alteration in levels of superoxide upon the treatments.

### 2.6. ROS Levels Measured with CellROX Green 

ROS can come from many areas in the cell. CellROX green is another fluorogenic probe used for measuring oxidative stress in live cells, reporting levels of ROS including superoxide and hydroxyl radicals. ML-1 cells treated with 22.8 µg/mL of ZnO and 45.5 µg/mL of CuO had essentially no significant impact on levels of ROS both at 24 h (Figure 7A) and 48 h of incubation (Figure 7B) with these MONPs. After 24 h of MONP treatment, CA77 cells display no significant elevation of ROS (Figure 7C), whereas both ZnO (*p* < 0.01) and CuO (*p* < 0.001) treatments for 48 h (Figure 7D) in CA77 cells resulted in the elevation of oxidation stress.

### 2.7. Apoptosis

Although ML-1 cells (Figure 6) and CA77 cells showed no significant increase in ROS after 24 h treatment of ZnO and CuO, a potential mechanism behind the decreased cell viability (Figure 4 and Figure 5) that we saw in earlier XTT experiments would be apoptosis. To test this possibility, we tested both thyroid cancer cell lines with the above-stated IC_50_ values for ZnO and CuO (Figure 8A,B). ML-1 cells showed an increase both in early and late apoptosis with the increased average percentage of cells in either phase of apoptosis (Figure 8A, *p* < 0.01), although ZnO-treated cells were not statistically significantly different from the non-treated control group due to a high standard error (Figure 8A). Interestingly, in CA77 cells, the majority of apoptosis cells with MONPs are in an early stage of apoptosis (Figure 8B, *p* < 0.0001), while ML-1 cells are under a late apoptotic phase in the same treatment. Together, both cells display an elevated induction of apoptosis within 24 h of treatment with these MONPs.

### 2.8. Altered Genome with ZnO/CuO Treatment upon ML-1 Cells 

RNAseq data were aligned with the human genome, which resulted in a total of 19,000 differentially regulated genes when treated with ZnO or CuO in ML-1 cells. However, only 271 upregulated genes and 327 downregulated genes from the ZnO-treated cells were not excluded by a two-fold cut-off threshold and a *p*-value less than 0.05. Our GO term analysis revealed that the most frequently upregulated pathways include p53, Wnt signaling, CCKR, and Gonadotropin hormone receptor pathway (Table 1, the column labeled as “Up by ZnO”). A few of the most highly upregulated genes, with at least a 20-fold increase in their transcripts, include metal-response genes (*MT2A* and *ZNF100*) and genes involved in the p53 pathway (*MYC* and *CCNA2*). Interestingly, cell-cell adhesion, inflammation, and Wnt signaling pathways were found to be the most frequently downregulated (Table 1, the column labeled “Down by ZnO”). Consistently, our analysis displayed the most significantly downregulated genes (more than a 20-fold decrease in their transcripts) with ZnO treatment, including *CDH16*, a gene implicated in the cell adhesion pathway. In addition, *CYP17A1 and CYP2E1,* implicated in cytochrome p450 monooxygenase metabolism, were significantly downregulated in our analysis.

After CuO treatment, the most frequently upregulated pathways include Wnt, gonadotropin-releasing hormone, and CCKR pathway (Table 1, the column labeled “Up by CuO”). Among many, the following six transcripts were most significantly upregulated (more than 20-fold): *POLE, PCNA, DUSP5, SPRY1, PTPRK*, and *MTUS1*. In particular, *POLE* and *PCNA* are implicated in DNA replication, while *MTUS1* is known to be involved in cell proliferation. The most frequently downregulated pathways include Wnt, inflammation, and cadherin signaling pathway (Table 1, the column labeled “Down by CuO”). Genes that were most significantly downregulated (more than 20-fold) included *PCDH15, GNAT1, TENM2, DYRK2, HMOX1*, and *KIAA1001*. It was found that *PCDH15* and *TENM2* are known to be associated with cell adhesion regulation, consistent with the most frequently downregulated pathways. Together, it was found that genes implicated in DNA replication were highly upregulated, while genes whose functions are implicated in cell adhesion are significantly downregulated. 

### 2.9. Altered Genome with ZnO/CuO Treatment on CA77 Cells

CA77 cells treated with ZnO NPs had 256 genes upregulated and 136 genes downregulated. The most frequently upregulated pathways include the thyrotrophin-releasing hormone receptor signaling pathway, inflammation, and Wnt signaling (Table 1, the column labeled “Up by ZnO” in CA77 cell). The three most highly upregulated genes were found to be *Lmx1b* (regulating protein binding to DNA), *Ccr5* (inflammation), and *Rtkn2* (apoptotic signaling). Downregulated pathways after ZnO NP treatment include integrin signaling pathways, inflammation, and cytoskeletal regulation (Table 1, the column labeled “Down by ZnO” in CA77 cells). The three most severely downregulated genes were found to be *Bad* (apoptosis), *Nlrp10* (immune response), and *Bag6* (autophagy). Together, this result suggests that both inflammation and apoptotic mechanisms are not balanced with the ZnO treatment in CA77 cells.

CA77 cells treated with CuO NPs expressed 118 upregulated genes and 163 downregulated genes. Most of the upregulated genes were not distinctive biological pathways. There were only three main upregulated pathways: TGF-beta signaling, interleukin signaling, and inflammation mediated by chemokine and cytokine signaling pathways (Table 1, the column labeled “Up by CuO” in CA77 cells). The three most highly upregulated genes were found to be *Aox4* (Xenobiotic function), *Dpp4* (collagen binding), and *Pdcd1Ig2* (cytokine regulation). In contrast, inflammation mediated by chemokine and cytokine signaling pathways, angiogenesis, and VEGF signaling were the most frequently downregulated pathways affected by CuO treatment (Table 1, the column labeled “Down by CuO” in CA77 cells). The three most severely downregulated genes in our analysis were *Stx11* (interferon signaling), *Frat1* (Wnt signaling), and *Pglyrp4* (humoral immune response).

## 3. Discussion

The present study provides novel insights into the molecular mechanisms underlying MONPs-mediated toxicity. Given the large quantity of MONPs discharged after their use, all organisms from prokaryotes to the most complex eukaryotes have been exposed to these materials. The toxicity of MONPs was addressed through many different methods in this study including cell viability, ROS, apoptosis, and RNAseq. To start, yeast, ML-1, CA77, and fibroblast cells were all used to quantify cell viability. Initially, yeast cells were treated for 24 h with TiO_2_, SnO_2_, ZnO, or CuO. There was no significant viability decrease in yeast cells (Figure 3), which is substantially different from mammalian viability trends (Figure 4 and Figure 5). The structure of yeast compared to mammalian cells is very different. One of the main distinctions is the cell wall component, which could be a major factor as to why the MONPs are not hurting growth.

Most current literature focuses on cervical, breast, and lung cancers but investigating human thyroid cancer (ML-1) can be a unique discovery in the field. Not only are ML-1 cells rarely studied but comparing toxic effects in human thyroid cells with those in rat thyroid cells can give more insight into which MONPs exert more toxicity. Finally, we used fibroblast cells to check if these NPs can be used as alternative cancer therapy as previously discussed in the literature [16] One of the major conundrums in our study is ROS results. Based on recent literature [17,18], increased ROS is the main cause of decreased cell viability after MONP treatment. Unlike the usual trend, we saw insignificant ROS production consistently in both thyroid cancer cell types for the first 24 h after treatments. Because of this result, we decided to use two different methods to quantify ROS (DHE and CellROX green) to confirm our results. DHE experiments showed insignificant changes in ROS with both cell lines (Figure 6) after 24 h, 4 h, or even 30 min. DCF was used on ML-1 cells to reevaluate levels of ROS, but the same trend was evident after 24 h (Data not shown). Interestingly, CA77 cells displayed an increased level of ROS at 48 h (Figure 7), though ML-1 cells showed no increased ROS, leading us to believe there is a different mechanism in place behind decreased cell viability after treatment. Kalhson et al. demonstrated alternative mechanisms behind cell death upon MONP treatment. For example, excess Zn^2+^ ions can cause nonapoptotic cell death by inhibiting ATP synthase [19]. Interestingly, copper toxicity is also linked to altered mitochondrial function [19] through protein aggregation at the mitochondria, thus most likely impairing the energy production process occurring in the organelle. Another unique contribution of the present study is that the RNAseq analysis offers a plethora of novel explanations for non-ROS-mediated pathways that lead to cell death. Accordingly, the present study proposes that ZnO affects several pathways of the cellular stress response, which leads to cell death. Similarly, we concluded that differentially expressed genes implicated in the DNA replication process and inflammation in CuO-treated cells are the main mechanisms of cell death.

### 3.1. Both ZnO and CuO Lead to a Reduction in Cell Viability

Lines of research, in which a range of cancer cells have been used for their research, have suggested MONPs as anticancer therapy agents [16]. Although cancer cells are surrounded by non-cancerous cells in vivo, many published papers fail to include normal tissue cells as a control experiment in their research. For example, Bai and colleagues reported that MONPs induce a significant level of cytotoxicity in cancerous cells without mentioning the effects of their impact on regular tissue cells [20]. To elucidate the impact of MONPs in normal tissue cells, our study included mouse fibroblast cells along with two cancer cell lines, ML-1 and CA77. One interesting finding from our analysis is that ZnO NPs caused significant viability defects in the fibroblast cells (Figure 5C), while CuO-treated cells demonstrated no significant viability defect. Consistently with our ZnO-treated viability defects, Teng Ng and Yong used human fibroblast cells (MRC5) treated with ZnO NPs and observed a significant dose-dependent viability decrease [21]. Together, it is considered that ZnO NPs make them a poor candidate for cancer therapy. Out of the four MONPs (SnO_2_, TiO_2_, ZnO, and CuO), ZnO and CuO only had obvious toxic effects when treated to ML-1 cells, CA77 cells, and mouse fibroblast cells by showing significantly decreased cell viability. 

ML-1 cells represent human thyroid cancer, which is an excellent model for developing how MONPs function in vitro. ML-1 cells are very rarely researched compared to other cancer cell lines including CA77 cells. In our XTT viability experiment, ML-1 cells showed viability defects with ZnO concentration as low as 40 µg/mL (Figure 5A). CuO-treated ML-1 cells showed viability defects starting at 60 µg/mL of CuO (Figure 5A). From these results, IC_50_ values were calculated to be 22.4 µg/mL for the ZnO-treated culture and 45.5 µg/mL for the CuO-treated culture. Although we found that IC_50_ values calculated in several reports were significantly different depending on the cell lines used, ranging from 10 to 100 µg/mL of ZnO. However, a few reports, including Namvar et al. that used mouse breast cancer (4Ti) treated with ZnO NPs, revealed their corresponding IC_50_ values very close to the IC_50_ values we reported here [22]. Another popular cancer cell line, MOLT4, was evaluated with CuO NPs and produced an IC_50_ value of 38.41 µg/mL [23]. Given the comparably low IC_50_ values in ML-1 cells as compared with other sensitive cell lines (4Ti and MOLT4) to ZnO and CuO, we came to the conclusion that ML-1 cells can be categorized as a highly sensitive cell line to these MONPs. 

Evaluating the viability of CA77 cells can give further insight into animal models affected by ZnO and CuO treatment. CA77 cells treated with CuO and ZnO have a similar decreased cell viability pattern to ML-1 cells (Figure 5B). However, CA77 cells appear to be more resistant to these MONP treatments based on their higher IC_50_ values compared to those of ML-1 cells. After ZnO and CuO treatment, 50% of cells were inhibited in viability at 68.9 µg/mL and 72.9 µg/mL, respectively (Figure 5B). To compare, B9 rat cancer cell lines were used by Kukia and Abbasi and treated with CuO MONPs. B9 cells displayed significantly lower IC_50_ values (12.01 µg/mL) after CuO NP treatment, making them very susceptible to MONPs [24]. Varying IC_50_ values are expected after the treatment depending on how different cancer cells react. Based on our results, CA77 cells showed decreased cell viability even with a higher concentration of ZnO and CuO than ML-1, indicating that CA77 cells are more resistant than ML-1 to these MONPs.

### 3.2. Levels of Reactive Oxygen Species Were Not Altered between Treatments with IC_50_ Values

Elevated levels of reactive oxygen species (ROS) can often lead to decreased cell viability as previously shown [25,26]. ROS are natural byproducts of cellular oxidative metabolism and play an important role in cell survival, cell death, differentiation, cell signaling, and inflammation-related factor production [27]. Our ROS experiments were inconsistent with current literature trends. Both ML-1 and CA77 cell lines (Figure 6) treated with ZnO and CuO for 24 h showed essentially no changes in the levels of superoxide. The modestly altered levels of ROS between treatments were not statistically significantly different from one another (Figure 6). Similarly, after 4 h or 30 min of treatment, the superoxide production level in both ML-1 and CA77 cell lines was similar to the level of superoxide in non-treated control cells (Figure 6). We hypothesized that superoxide in our experiments could have been elevated as soon as we treated cells with these MONPs. However, ROS elevation was not observed even 30 min after the treatment of CuO and ZnO (Figure 6). Therefore, our finding is not consistent with a 2020 paper that evaluated total levels of ROS in Gingival squamous cancer cells (GSCC) [27], including superoxide, and that discovered elevated superoxide upon short-term exposure to ZnO MONPs (30–60 min). When it comes to the treatment of CuO, Bondarenko et al. detected superoxide using DHE and concluded that among MONPs tested, CuO was one of the greatest superoxide producers [28]. Together, although many reports have argued that increased ROS is the mechanism behind decreased cell viability, the observation of cell viability defects in our experiments with ZnO and CuO does not seem to be due to elevated ROS. 

### 3.3. Apoptosis Was Induced Both in ML-1 and CA77 Cells with MONPs

Apoptosis is the process behind programmed cell death. No discernable alteration in ROS levels (Figure 6) with viability defects (Figure 5) in response to MONPs (24 h treatment) suggests that non-ROS-mediated apoptosis could be the main factor behind the observation. In light of the finding of increased late apoptosis in ML-1 cells and increased early apoptosis in CA77 cells after 24 h of incubation with the IC_50_ values of ZnO and CuO MONPs (Figure 8) points to ML-1 being less resistant to the treated MONPs. Other researchers investigated non-ROS-mediated apoptosis with ZnO NPs. They found that the dissolution of Zn^2+^ ions from ZnO MONPs can lead to an imbalance in homeostasis, then damage to lysosomes and mitochondria, and finally cell death [29]. ZnO MONPs are popularly used as drug delivery systems that can be designed to bind to specific receptors on the membrane. The binding of these nanomaterials to death receptors may lead to cell death by inducing a signal complex, which cleaves caspase 3 and activates caspase 8 [30]. Our current study emphasizes non-ROS mediated apoptosis after ZnO and CuO treatment to both cell lines (ML-1 and CA77).

### 3.4. Upregulated Transcripts Implicated in p53 and Wnt Pathways after ZnO Treatment of ML-1 Cells 

After analyzing differentially expressed genes (DEGs) in ZnO-treated ML-1 cells, p53 and Wnt signaling pathways were not only frequently (Table 1) but also highly upregulated (Figure 9). Two of the most upregulated genes involved in the p53 pathway include *MYC* and *CCNA2*. It is well-known that the p53 pathway is activated in response to stress signals like DNA damage [31]. Given that a group of researchers recently found that ZnO (50 µg/mL) induces DNA damage in A549 (human lung carcinoma) cells [32], we conjecture that the ZnO treatment in our experiments might have resulted in DNA damage, which then upregulated the transcripts in the p53 pathway. It appears that a similar DNA damage response occurs as treated with ZnO in different organisms. For example, in a recent study, *CCNA2* was also upregulated in *D. melanogaster* after ZnO treatment [26]. The upregulation of the p53 pathway in ZnO-treated cells is highly consistent with the increased late apoptosis observed in our apoptosis results (Figure 8).

In addition, *SMAD5* and *CDH1* are implicated in the Wnt signaling function that regulates cell growth, differentiation, and cell death. *SMAD5* was found to be a tumor suppressor candidate [33] and *CDH1* is a tumor suppressor, too [34]. Both genes were upregulated based on our RNAseq analysis after ZnO treatment in ML-1 cells (Table 1). Similarly, it was found by Salesa et al. that *CDH1* was also upregulated in HaCaT cells after ZnO treatment [35]. The upregulation of these tumor suppressor genes in our experiments may likely be correlated to low cell growth in ZnO-treated cells due to the inhibition of cell proliferation, as backed up by our XTT viability analysis (Figure 4 and Figure 5A).

### 3.5. Severely Downregulated Genes Include Inflammation and Cadherin Signaling after ZnO Treatment in ML-1 Cells 

The most severely downregulated gene discovered in the ZnO-treated ML-1 cells are connected with cadherin signaling (*CDH16*) and xenobiotic metabolism (Figure 9). Consistent with our study, a 2020 study evaluated ZnO NPs in human tenon fibroblast cells (HTFs) and found downregulation of the E-cadherin pathway [36]. Given cadherin plays a role in the growth and cell-cell adhesion, the downregulation of *CDH16* suggests that ML-1 cells treated with ZnO NPs could lose cell-cell adhesion and growth properties. 

### 3.6. Genes Implicated in DNA Repair and Growth Inhibition Were Significantly Upregulated after CuO Treatment to ML-1 Cells 

We found that gene transcripts implicated in growth inhibition were significantly upregulated: *SPRY1, DUSP5*, and *MTUS1* (Figure 10). In particular, *MTUS1*, a known tumor suppressor involved in Wnt signaling [37], and therefore, the upregulation of it is correlated with an inhibition of cell growth, which we observed in Figure 4 and Figure 5. 

DNA repair pathway genes were found to be significantly upregulated in our experiments: *PCNA* and *POLE,* (Figure 10). Since lines of investigation including Martinez et al. point to DNA damage by the presence of CuO in cells [23], the upregulated DNA repair pathway reflects DNA damage in response to CuO treatment in the present study. One interesting, downregulated gene shown in our model (Figure 10) is *HMOX1,* which is known to act against oxidative stress [38], suggesting no elevation of ROS in our CuO treatment experiments. Therefore, this RNAseq result of the low transcript of *HMOX1* is consistent with our non-ROS-mediated cell growth defects.

### 3.7. Genes Implicated in Wnt Signaling and Cadherin Pathways Were Severely Downregulated after CuO Treatment to ML-1 Cells

Downregulated pathways after CuO treatment are very similar to downregulated pathways with ZnO treatment to ML-1 cells. For example, Wnt signaling (*WNT7B*) is downregulated (Figure 10). As mentioned previously, the downregulation of Wnt pathways can lead to inhibited proliferation [39]. Therefore, the growth defect in ML-1 cells in response to CuO is in part due to the downregulation of cell proliferation by Wnt signaling. In addition, both *PCDH15* and *TENM2*, associated with cadherin signaling, were also downregulated in our experiment (Figure 10). Therefore, it is considered that the downregulation of adhesion signaling might further exacerbate the growth potential of ML-1 cells in the presence of CuO.

### 3.8. Inflammation Is the Most Upregulated Pathway after CuO Treatment to CA77 Cells 

After CuO MONP treatment to CA77 cells, inflammation was the most frequently upregulated pathway (Table 1). In addition, we found that *Dpp4,* a modulator of inflammation [40], was significantly (>20-fold) upregulated in CA77 cells with the present study (Figure 11). *Pcd1l2g* is another inflammatory gene upregulated in CA77 cells after treatment (Figure 11). It is clear that CuO MONPs cause increased inflammation as reported many times in the recent literature [41,42,43]. Given that CuO MONPs can bind to cell death ligands to induce non-ROS-mediated apoptosis [44], our RNAseq studies further confirm non-ROS-mediated apoptosis through increased inflammation pathways. 

### 3.9. Severely Downregulated Angiogenesis and Cadherin Pathways after CuO-Treatment to CA77 Cells

After CuO treatment, the few, most severely downregulated genes shown in Figure 11 are *Clec9a* and *Il10rb* (cytokine pathway), and *Frat1* (Wnt signaling). In particular, *Frat1* mainly functions as a positive regulator of Wnt signaling and regulates tumor progression [45]. Given that Wnt signaling is widely known for promoting cell proliferation, our data on the severe downregulation of the Wnt signaling factor suggest growth defects in the presence of CuO.

### 3.10. Genes Involved in Inflammation Pathways Were Significantly Upregulated after ZnO Treatment to CA77 Cells

Genes involved in inflammation were found to be upregulated (more than a 20-fold increase) after ZnO MONP treatment in CA77 cells: *Ccr5* and *Zfp36l1* (Figure 12). It is known that ZnO MONPs induce inflammation as reported many times, especially in mammalian cells [46,47,48]. Therefore, in CA77 cells, it appears that *Ccr5* and *Zfp361l* upregulation in response to ZnO treatment takes place to promote inflammation. 

### 3.11. Inflammation and Integrin Genes Were Severely Downregulated after ZnO Treatment to CA77 Cells 

*Ackr3,* associated with inflammation, was downregulated severely after ZnO treatment to CA77 cells (Figure 12). Anti-inflammatory mechanisms were discussed in a paper by Agarwal and Shannmaug [49]. They explained that ZnO NPs used for drug delivery functions can elicit anti-inflammatory properties like stability characteristics.

Integrin is involved in cell-cell and cell-extracellular matrix adhesion [50], and we found that integrin pathways are severely downregulated after ZnO NP treatment to CA77 cells (Figure 12). *The cd2* gene is implicated in the integrin pathway and was downregulated significantly (Figure 12). From our current studies, 24 h of treatment causes CA77 cells to detach from the bottom of the flask. Additionally, Fernández and colleagues evaluated ZnO MONPs and found a similar result [51].

## 4. Materials and Methods

### 4.1. Materials and Reagents Used in the Study

The following MONPs were purchased from US Research Nanomaterials Corporation (Houston, TX, USA): ZnO, CuO, SnO_2_, and TiO_2_. ML-1 (human thyroid cancer) cells were purchased from DSMZ, German Collection of Microorganisms and Cell Cultures (Braunschweig, Germany). CA77 (rat thyroid cancer) cells were obtained from Jordan Valley Innovation Center (JVIC) from Dr. Paul Durham (Missouri State University). Mouse fibroblast cells were obtained from Dr. Chris Lupfer (Missouri State University). DMEM (Dulbecco’s Modified Eagle Medium), DMSO (Dimethyl Sulfoxide), Dihydroethidium (DHE), CellROX green, FBS (Fetal Bovine Serum), TRIzol, PI (Propidium Iodide), Annexin-V-APC, and XTT kit were obtained from FisherSci (Waltham, MA, USA). All other reagents and plasticware used in the study were purchased from FisherSci unless otherwise stated.

### 4.2. ZnO and CuO NP Scanning Transmission Electron Microscopy (STEM)

These MONPs (1 mg/mL) were received as they are, dispersed in water. Only ZnO and CuO caused a cell viability decrease (see Section 2), and therefore, we decided to proceed with just these two nanoparticles for the following experiments. A JEOL 7900F scanning electron microscope (SEM) (Peabody, MA, USA) with a scanning transmission electron microscopy (STEM) detector was used to image the CuO and ZnO nanomaterials. For each of the nanoparticle materials, CuO (40 nm) and ZnO (35–45 nm), the sample preparation procedure was identical. Each nanomaterial sample was weighed out to approximately 10 mg with a 0.01 mg resolution Ohaus Discovery scale. The 10 mg of nanomaterial was added to 10 mL of 18 MΩ deionized water from a Milli-Q IQ 7000 series ultrapure deionized water polisher in a 20 mL scintillation vial. The dispersion was sonicated with a Sonics VCX 750 immersion sonicator using a cup horn accessory. The sonication bath was maintained at 10 °C using a chiller and sonication was conducted for 30 min at 60% power in pulse mode on/off: 10/5 s. The nanomaterial dispersion was diluted to a concentration of 0.1 mg/mL and subsequently sonicated again for 10 min using the same sonication conditions as the stock dispersion. After dilution, 20 µL of each nanomaterial dispersion was drop cast onto a Pelco carbon film on a 300-mesh copper TEM grid and allowed to air dry in a desiccator box for 4 h. The TEM grid was placed on an SEM holder and placed in the SEM chamber and vacuumed until a pressure of 9.6 × 10^−5^ Pa was reached. Imaging was conducted in the STEM mode at 30 kV acceleration voltage, and a working distance of 4.4 mm. Images were taken at 25 and 50 kX.

### 4.3. X-ray Diffraction of ZnO and CuO NPs

X-ray diffraction (XRD) was conducted using a Bruker D2 phaser XRD system (Bilerica, MA, USA) to characterize the crystallinity of the CuO and ZnO nanomaterials. Both nanomaterials were prepared similarly, wherein 10 mg of each was weighed on an Ohaus Discovery scale with 0.01 mg resolution and added to 3 mL of methanol in a 5 mL glass crucible and stirred until a thick paste was formed. For XRD, specific mass/volume of dispersion or paste is not critical if all compared materials are prepared in the same way. The paste was then coated onto a zero-background silicon plate and dried on a hotplate at 75 °C for 10 min. The zero background silicon plate with dried nanomaterial was placed in the XRD chamber. The conditions for the analysis were set to X-ray radiation of 30 kV from a CuKα X-ray source with a theta range of 20–90° (2θ), at 0.02° (2θ) steps and a sample rotation set to 10 rpm.

### 4.4. Yeast Culture and Viability Assay

The budding yeast *Saccharomyces cerevisiae* strain S288C was grown in 2X synthetic-defined glucose (2X SD + Glu) media and placed in a shaking incubator at 30 °C to grow overnight as described previously [52,53,54]. The optical density (OD) was taken of the liquid cell culture with a spectrophotometer at 600 nm and recorded to be used in the cell stock calculation. The cell stock was then diluted to 0.2 OD at 600 nm. Each of the nanomaterials used, ZnO, CuO, TiO_2_, and SnO_2_, was independently suspended in water at a concentration of 1000 µg/mL. 

For yeast viability experiments, a round-bottom 96-well plate was utilized to get the most accurate OD reading at 600 nm. A 1:2 serial dilution was performed of the nanoparticle stock solution to provide treatment conditions ranging from 0 µg/mL to 100 µg/mL. Yeast cells (OD of 0.2 at 600 nm) were then added to only the first three rows of the seeded wells leaving the bottom three rows to be a comparative control only containing nanomaterial and 2X SD + Glu media. Each of the treatment concentrations and their corresponding control was seeded in a triplicate manner on the 96-well plate. The seeded plate was then placed in a BioTek ELx808 Absorbance Microplate Reader where the OD was read at 594 nm every 30 min over a period of 24 h. This experiment was conducted multiple times. The data shown (Section 2.3) are representative of the reproducible data collected. 

### 4.5. Statistical Analysis for Yeast Viability Experiments

The OD obtained from the treated wells at 594 nm after 24 h incubation was subtracted from the corresponding control wells without cells to obtain a corrected OD (the equation used for this calculation was as follows).
Net cell density (OD) = OD^C^ − OD^M^.
where OD^C^ is the OD value at 594 nm from a culture well with cells and media and OD^M^ is the OD value at 594 from a culture well with media and nanoparticles only.

From each triplicated treatment, we determined the mean corrected OD in order to create growth curves. GraphPad Prism9 was used to perform a one-way ANOVA statistical analysis to determine if the mean OD of treated samples had a statistically significant difference from that of the non-treated samples. Endpoint graphs were made using the OD at the 24 h mark and asterisks were used to identify values significantly different (*p*-value < 0.05). Dunnett’s multiple comparisons were used to visualize the variance in control and treatment groups. The graph bars represent an average of three replicates and the error bars represent the standard deviation. Statistically significant data are represented as * *p* < 0.05, ** *p* < 0.01, *** *p* < 0.001, **** *p* < 0.0001. The obtained results were validated by repeating the experiments in triplicate.

### 4.6. Mammalian Cell Cultures

ML-1, CA77, and fibroblast cells were maintained in DMEM with 10% FBS, 5% penicillin, and 5% amphotericin. These cell lines were cultured in a CO_2_ incubator with 95% O_2_ and 5% CO_2_ as previously described [55,56,57]. 

### 4.7. XTT Cell Viability Using Mammalian Cells 

An XTT assay was performed on mouse fibroblast, ML-1, and CA77 cells to explore the impact of ZnO, TiO_2_, CuO, and SnO_2_ on a flat bottom 96-well plate that was used for cell culture for the cells to adhere to the bottom of the plate. In the 96-well plate, 10,000 ML-1 cells were seeded per well and incubated in a 37 °C incubator for 24 h. On day two, DMEM media was removed from each well, and differing amounts of MONPs, ranging from 4 to 100 µg/mL, were treated in triplicate via serial dilution. A positive control, DMSO, was also done in triplicate with concentrations of 5%, 10%, and 20%. Treated cells were incubated for another 24 h. After incubation, a 1:200 XTT solution was prepared to contain XTT buffer and XTT activation reagent. Each well was treated with 25 µL of the prepared XTT stock for 7 h at 37 °C. The BioTek ELx880 Absorbance Microplate Reader (Winooski, VT, USA) was used to measure the amount of reduced formazan dye formed in each well. Absorbance values at 450 nm were subtracted by absorbance values at 630 nm to gain a net colorimetric change that most accurately represents the cell viability in the culture.

### 4.8. Calculation of IC_50_ Value

The AAT Bioquest website was used (https://www.aatbio.com/tools/ic50-calculator, accessed on 1 July 2022) to determine IC_50_ values representing the concentration of MONPs at which 50% of cultured cells are inhibited in cell viability. Data from ML-1 cells were inserted into the equation provided on the webpage to determine IC_50_ values; it was determined that ZnO’s IC_50_ was 22.8 µg/mL and CuO’s IC_50_ value was 45.5 µg/mL. The IC_50_ value was then calculated for CA77 cells from XTT cell viability data. These values were as follows: 67.98 µg/mL for ZnO and 72.88 µg/mL for CuO. 

### 4.9. Measuring Levels of Reactive Oxygen Species

Two separate 24-well plates were seeded with 50,000 ML-1 and CA77 cells per well and placed into an incubator for 24 h at 37 °C. The following day, the cells were incubated with either ZnO or CuO at its concentration of IC_50_ value: 22.8 µg/mL of ZnO and 45.5 µg/mL of CuO. These treatments were conducted in triplicate as well as the positive control (DMSO) treated at 20%, and the untreated control. An untreated control group and two blanks (no Dihydroethidium, DHE) were only treated with DMEM. The third day consists of washing treated cells once with 500 µL 1X phosphate-buffered saline (PBS). To detach the cells, 250 µL trypsin + EDTA was added to each well and then placed in the incubator for 15 min. After the incubation period, trypsin+EDTA was neutralized with 500 µL pre-warmed DMEM. The cell suspension in each well was transferred to labeled 2 mL centrifuge tubes to be centrifuged for 10 min at 400× *g*. During this time, the ROS indicator, DHE (dihydroethidium), was prepared by mixing 26 µL DHE into 26 mL 1X PBS in a 50 mL Falcon tube. Given DHE’s light sensitivity, the 50 mL Falcon tube containing indicator mix was covered with aluminum foil. The supernatant from the centrifuge tubes was removed and 1 mL of indicator mix was added to each tube except the two blanks. Instead, the two blanks received 1 mL 1X PBS. After pipetting up and down, each resuspended mix was transferred to new foil-covered 2 mL centrifuge tubes. The transferred cells were then incubated for 1 h before being analyzed by an Attune NxT Flow Cytometer. The dyed samples were measured at 518/606 nm.

### 4.10. CellRox Green Reactive Oxygen Species

Three separate 18-well plates (Ibidi, Grafelfing, Germany) were seeded with 10,000 ML-1 cells per well. Each 18-well plate was incubated for different periods of time, including 24 and 48 h. After 24 h of incubation, all plates were treated with the IC_50_ concentration of either ZnO or CuO. Four treatment groups were conducted in triplicate: NTC, ZnO, CuO, and 10% H_2_O_2,_ where H_2_O_2_ was used as a positive control. After treatment, the cells were washed with 100 µL of 1X PBS, and then 5 μM of cellROX green dye solution was applied to each well. Cells were then washed three times with a live cell imaging solution (Fisher) to make sure phenol red from DMEM does not interfere with imaging. Immediately after, cells were visualized with a confocal microscope. We chose 10 cells with detectable fluorescence randomly to evaluate the fluorescence intensity with ImageJ. 

### 4.11. Apoptosis Measurement

For this experiment, an Annexin-V binding buffer and two dyes, Propidium Iodide and Annexin V-APC, were used. The 10x Annexin V binding buffer was made by combining 0.1 M of HEPES, 1.5 M sodium chloride (NaCl), 25 mM of calcium chloride (CaCl_2_), and molecular-grade sterile water. A 1x Annexin V binding buffer solution was made by diluting 18 mL of PBS and 2 mL of the 10x Annexin V binding buffer. Initially, 50,000 ML-1 or CA77 cells were seeded per well in a 24-well plate and then incubated for 24 h. On day two, MONPs were treated in concentrations of 22.8 µg/mL of ZnO and 45.5 µg/mL of CuO in triplicate along with DMSO and an NTC group. On day three, each well was washed with 300 µL of 1X PBS and then 250 µL of trypsin (25%, 2.21 mM) without EDTA was pipetted into each well. The plate was placed in an incubator for 20 min while the cells detached. To neutralize the non-EDTA trypsin, 300 µL of fresh DMEM was added into each well and then pipetted up and down. The treatment groups, NTC, and DMSO groups were resuspended in 100 µL of 1x Annexin V binding buffer solution, 5 µL of Annexin V-APC, and 5 µL of propidium iodide (50 µg/mL), then incubated in the dark for 30 min. After the incubation, an additional 400 µL of Annexin V binding buffer solution was added to each sample. The final analysis of each sample was based on the excitation properties of the dyes. Annexin V emits at 660 nm with a red laser and propidium iodide emits at 617 nm with a blue laser. Apoptosis experiments were conducted in triplicate and repeated three independent times. 

### 4.12. Total RNA Extraction

Extracting RNA requires four different phases: homogenization, separation, isolation, and washing according to the Invitrogen TRIzol protocol (Ambion, Carlsbad CA 92008). RNA extraction was conducted on CA77 cells as well as ML-1 cells. On the first day, 750,000 CA77 cells were seeded into each well of a 6-well plate. Two 6-well plates were used for a total of nine samples: three NTC, three 68.9 µg/mL of ZnO treated, and three 73 µg/mL of CuO treated. For ML-1 cells, ZnO at 22.8 µg/mL and CuO at 45.5 µg/mL were tested. After 24 h of incubation with the MONPs, total RNA was extracted. For this, 1 mL of TRIzol was added to homogenize the cells. Each sample was incubated for 5 min to complete the dissociation of the nucleoprotein complex. Next, each sample was incubated with 200 µL of chloroform for 2–3 min. After centrifugation, three phases formed: an upper colorless, an intermediate, and a lower red phase. The colorless phase contains RNA. The colorless phase was acquired, and then 100% isopropanol was added in equal parts. After another centrifugation, the supernatant was removed containing RNA, and 1 mL of 75% ethanol was added to each Eppendorf tube. Finally, ethanol was removed, and tubes were placed in the biosafety cabinet until completely dry. Thirty microliters of nuclease-free water was added to each tube to make a total RNA concentration of 2 µg and was shipped to Novogene for the production of cDNA libraries and Nextgen sequencing. 

### 4.13. RNA Seq Analysis 

After receiving sequencing files from Novogene, those were uploaded to *Basepair*. The spike-in ERCC was used when uploading files to reduce variability and identify errors. There was a linear correlation between the known concentration, expected concentration, and read counts for each of the different probes. Linear correlation indicates good sample quality. The first step in the RNA-seq pipeline was trimming to remove the poor quality of reads as well as adapter sequences. Each read was aligned to STAR, which is the reference genome to accurately identify the source of the gene. After all reads were aligned, Feature Counts were used for expression quantification to look at each gene and how many mapped reads are aligned to that specific gene. Next, the DESeq2 tool was used to conduct statistical analysis in the differential gene expression analysis step to find changes in gene expression between each sample group. After differential expression analysis, a volcano plot displays genes with a log10 *p*-value and a heat map was used to indicate up- and down-regulated genes. All up- and down-regulated genes were selected with a fold change of +2.01 and −2.01 with a *p*-adjusted value less than 0.05. The five most up- or down-regulated genes were chosen and investigated with IDTdna and Gorilla. CLC genomics (Qiagen) was used for CA77 cells to identify the differentially regulated genes when treated with ZnO and CuO. Datasets were downloaded, unzipped, then transported into CLC genomics. Fastq files were uploaded and then concatenated in Galaxy. After quality checks, RNA-seq analysis was conducted on each sample and CA77 gene expression was analyzed using CLC genomics. 

## 5. Conclusions

After analyzing our RNAseq data, many differentially regulated pathways overlapped after ZnO and CuO treatment in both cell lines (ML-1 and CA77). Briefly, both CuO and ZnO MONPs altered inflammation, cadherin, Wnt signaling, and p53 pathways in ML-1 cells. Similarly, CA77 cells exhibited altered Wnt, inflammation, and angiogenesis pathways after both treatments with ZnO or CuO. Therefore, animals and humans have similar pathways altered after ZnO or CuO treatment. Though these altered pathways are similar, the genes differentially regulated varied in each cell line. The biggest common denominators between both cell lines were found to be inflammation and Wnt signaling. These findings further confirm non-ROS-mediated apoptosis being the mechanism behind decreased cell viability in our experiments. Together, ZnO and CuO NPs cause dysregulation in inflammation, which could be the main factor behind increased apoptosis and decreased cell viability.

## Figures and Tables

**Figure 1 ijms-24-04055-f001:**
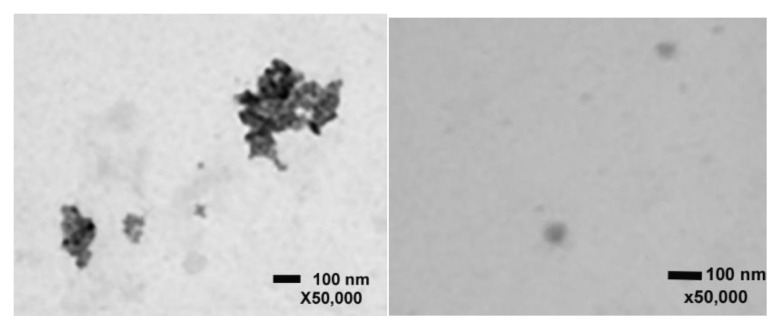
STEM images of CuO (**left**) and ZnO (**right**) at high (50,000×) magnifications. Bars on the images represent 100 nm.

**Figure 2 ijms-24-04055-f002:**
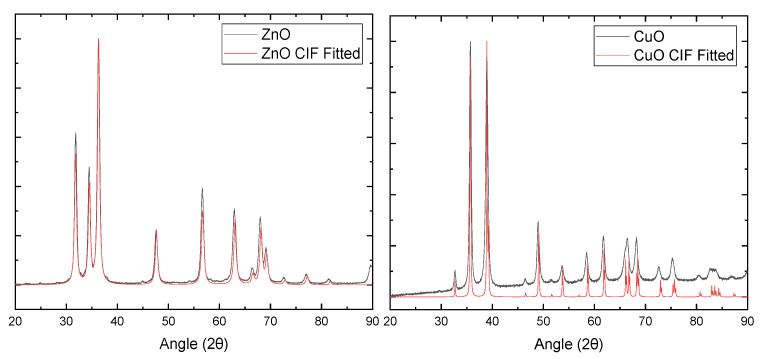
XRD plot for ZnO (**left**) and CuO (**right**). Red lines (ZnO CIF fitted and CuO CIF Fitted) include standard library peaks for ZnO and CuO, respectively. Black lines (ZnO and CuO) were obtained from our MONPs.

**Figure 3 ijms-24-04055-f003:**
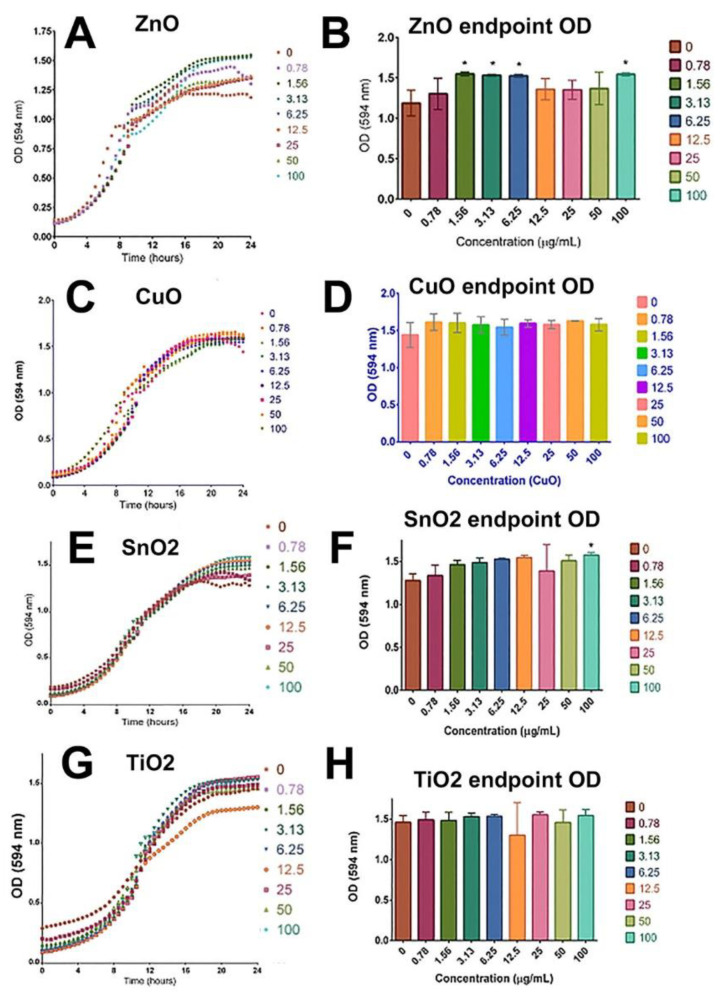
Cell viability measurement after 24 h of treatment. ZnO treatments (**A**,**B**). CuO treatments (**C**,**D**), SnO_2_ treatments (**E**,**F**), and TiO_2_ treatments (**G**,**H**). Asterisks were used to identify values significantly different. This experiment was repeated many times but one representative is shown. One asterisk (*) indicates *p* < 0.05.

**Figure 4 ijms-24-04055-f004:**
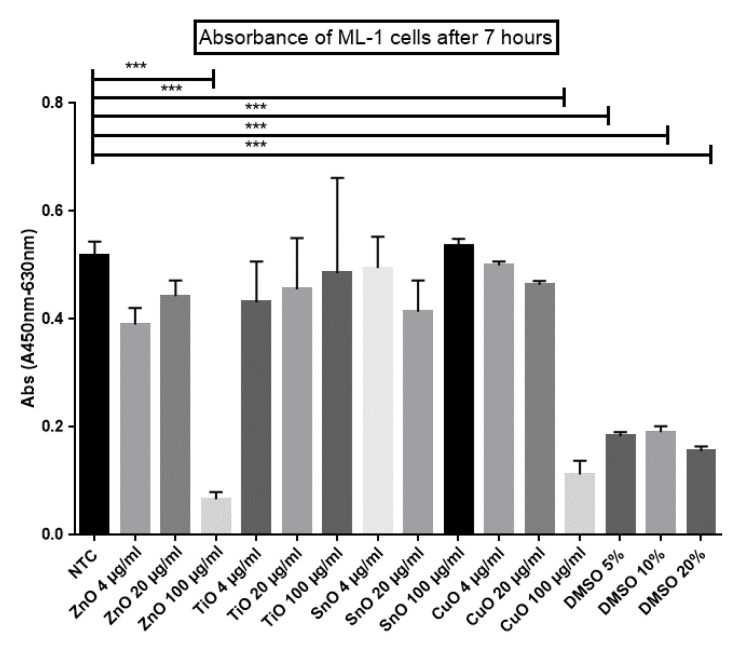
Effects of ZnO, CuO, TiO_2_, and SnO_2_ on ML-1 cell viability. DMSO serves as a positive control. Statistically significant results are indicated based on *p*-values: *** *p* < 0.001.

**Figure 5 ijms-24-04055-f005:**
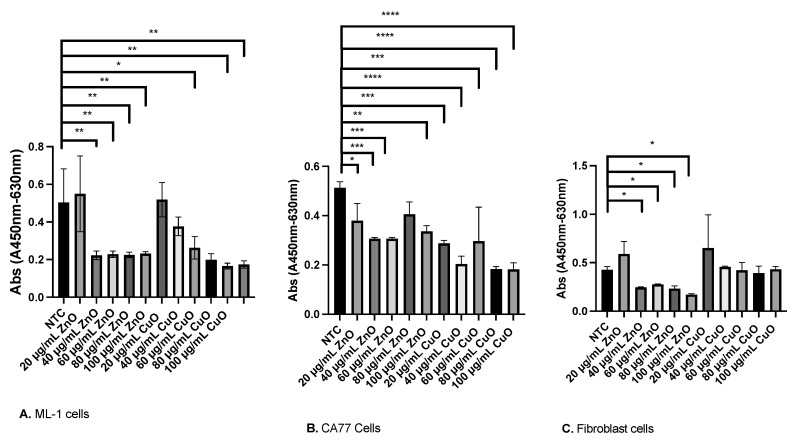
Minimal Inhibitory Concentration (MIC) for ZnO and CuO. (**A**) ML-1 cells exhibited decreased viability. (**B**) CA77 cell viability assay. (**C**) Viability of mouse fibroblast cells with various concentrations of ZnO and CuO. Statistically significant results are indicated based on *p*-values: * *p* < 0.05, ** *p* < 0.01, *** *p* < 0.001, and **** *p* < 0.0001.

**Figure 6 ijms-24-04055-f006:**
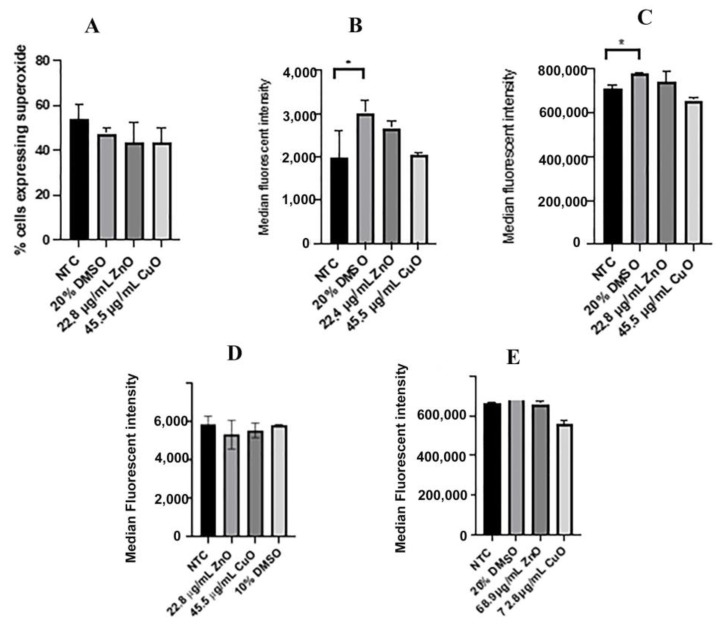
ML-1 and CA77 cells expressing superoxide production. (**A**) The percentage of ML-1 cells expressing superoxide after 24 h of treatment with 22.4 µg/mL ZnO or 45.5 µg/mL CuO. (**B**) Median superoxide production in ML-1 after 24 h. (**C**) Median superoxide production in ML-1 cells after 4 h. (**D**) Median superoxide production in n ML-1 cells after 30 min. (**E**) Median superoxide production in CA77 cells after 24 h. * indicates *p* < 0.05.

**Figure 7 ijms-24-04055-f007:**
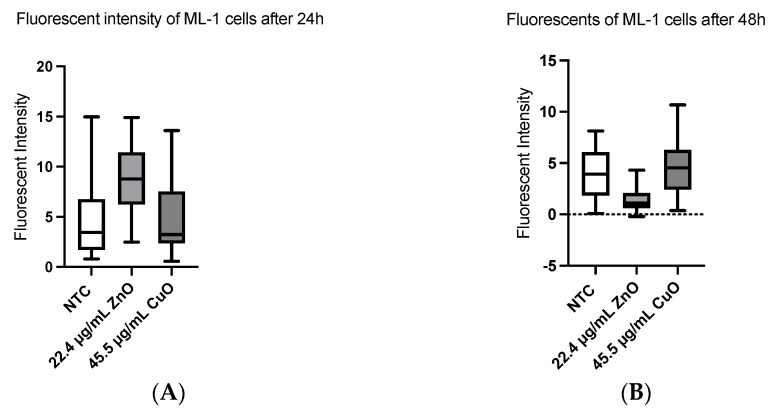
Levels of ROS measured by CellROX green probe. (**A**) ML–1 cells with the indicated amount of MONPs after 24 h. (**B**) After 48 h of incubation with the same amounts as shown in (**A**). (**C**) After 24 h of incubation with ZnO and CuO treatment on CA77 cells. Then 48 h of treatment (**D**). Two asterisks (**) and three asterisks (***) indicate *p* < 0.01 and *p* < 0.001, respectively.

**Figure 8 ijms-24-04055-f008:**
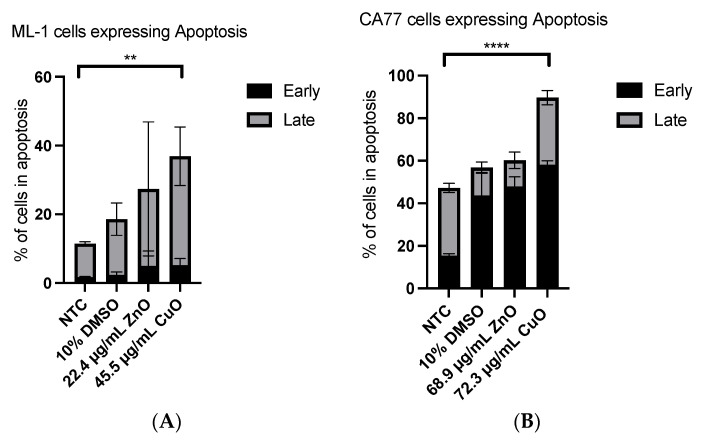
Quantitated apoptosis with two different cell lines: ML-1 (**A**) and CA77 (**B**). Two asterisks (**) and four asterisks (****) indicate *p* < 0.01 and *p* < 0.0001, respectively.

**Figure 9 ijms-24-04055-f009:**
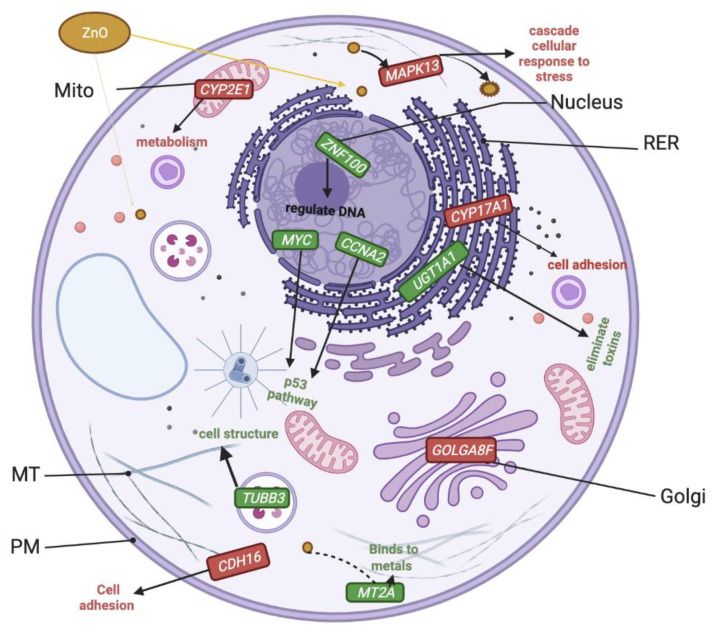
This schematic cell model presents the most highly upregulated (green) and severely downregulated genes (red) in ML-1 cells when treated with ZnO. The most frequently up- or down-regulated biological pathways in response to ZnO in ML-1 cells are displayed in Table 1. ZnO-treated cells resulted in differentially expressed genes (DEGs) involved in p53 (*MYC* and *CCNA2*), heavy metal response (*MT2A* and *ZNF100*), cell stress (*MAPK13* and *UGT1A1*), membrane trafficking (*GOLGA8F* and *TUB3*), metabolism (*CYP2E1*), and cell adhesion (*CYP17A1* and *CDH16*).

**Figure 10 ijms-24-04055-f010:**
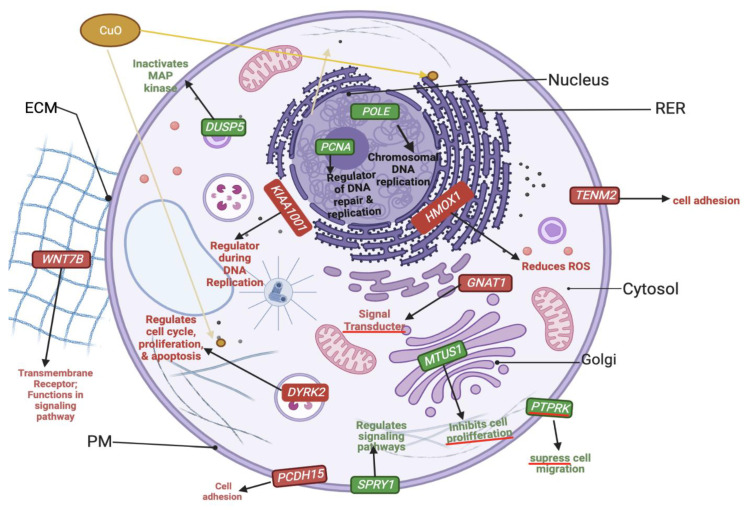
This model represents significantly upregulated genes (green) and severely downregulated genes (red) expressed in ML-1 cells when treated with CuO. Please see Table 1 for the most frequently up- and down-regulated pathways. As shown here, genes implicated in DNA repair (*POLE* and *PCNA*) or proliferation inhibition (*SPRY1, DUSP5*, and *MTUS1*) are highly upregulated (more than >20-fold), while genes implicated in Wnt signaling (*WNT7B*) or cell adhesion (*PCDH15* and *TENM2*) were significantly downregulated (>20-fold).

**Figure 11 ijms-24-04055-f011:**
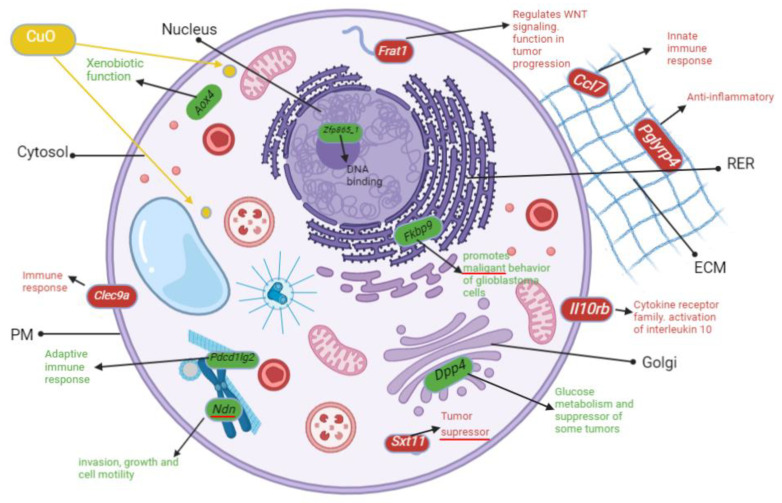
This model represents the most significantly upregulated (green) and severely downregulated (red) genes in CA77 cells with CuO. The most frequently altered pathways can be found in Table 1. In particular, genes implicated in inflammation (*Dpp4* and *Pdcd1Ig2*), xenobiotic function (*Aox4*), and transcription repressor (*Zfp865_1*) were upregulated more than 20-fold. In contrast, genes involved in Wnt signaling (*Frat1*), anti-inflammation (*Pglyrp4*), and cell growth (*Il10rb*) were severely downregulated.

**Figure 12 ijms-24-04055-f012:**
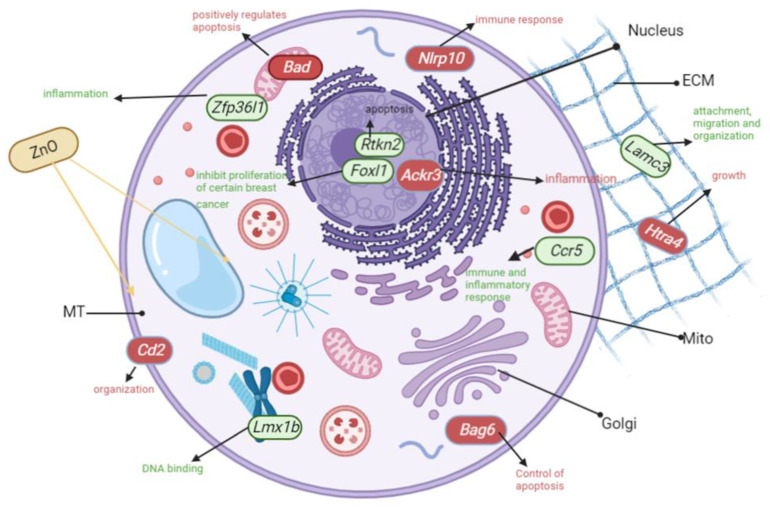
This model represents the most upregulated (green) and severely downregulated (red) genes expressed in CA77 cells after ZnO treatment. Table 1 contains biological pathways, most frequently up- or down-regulated.

**Table 1 ijms-24-04055-t001:** Altered pathways with differentially expressed genes (DEG) by ZnO or CuO in ML-1 and CA77 cells. Each well has a number of DEG implicated in the biological pathway listed in the column of “Pathways with DEG”. Altered pathways with high frequency are indicated in red; in particular, a large number of Wnt and inflammation signaling pathway genes are up- or down-regulated by ZnO and CuO.

Pathways with DEG	ML-1 Cells				CA77 Cells			
	Up by ZnO	Down by ZnO	Up by CuO	Down by CuO	Up by ZnO	Down by ZnO	Up by CuO	Down by CuO
Gonadotrophin-releasing hormone receptor	13	5	6	2	1			3
CCKR signaling	13	3	5		1	1		
Wnt signaling	12	7	6	7	3	2		
P53	12				1	2		
Nicotine degradation/nicotine acetylcholine receptor	11			6	2	1		2
Huntington disease	11	2	2		2	2		
Integrin signaling	10	5	4	4	2	4		
Inflammation	10	7		6	3	4	2	7
Apoptosis	9	2	3	2	1	1		
PDGF signaling	8		2		1	1	1	
Heterotrimeric G-protein signaling	8	6		10	6	2		3
Angiogenesis	8		4	3		2		
Presenilin (Alzheimer)	8		2	3	1	2		
Parkinson disease	7					1		
TGF-beta signaling	6		2	2			1	
Cadherin signaling	6	7		6		1		
Cytoskeletal regulation by Rho GTPase	5				3	2		
Glutamate receptor groups I, I, III	5			5	2			1
Ionotropic glutamate receptor	4							
FGF signaling	4	2		2	1	1		1
EGF receptor signaling	4	3			1	1		1
DNA replication	4							
Plasminogen activating cascade	4							
Blood coagulation	4	3			1			1
B cell activation	4	2				1		
Adrenaline and noradrenaline biosynthesis	3							
Endothelin	3							
Axon guidance	3							
PI3K kinase	3		2	2				
Oxidative stress response	3		2					
Notch Signaling	3		3		2			
Amyloid secretase		2						
Interleukin signaling			2	3			1	
Dopamine receptor				3	1	1		
Opioid				4	1			1
VEGF signaling					2	2		3
Thyrotrophin-releasing hormone receptor					4			
JAK-STAT					1	1		

## Data Availability

The data are contained within the article.

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
