# Peer review of "Non-ROS-Mediated Cytotoxicity of ZnO and CuO in ML-1 and CA77 Thyroid Cancer Cell Lines"

_ijms, 2023, doi:10.3390/ijms24044055_

Round 1

Reviewer 1 Report

Major Comments:

- The study is interesting as it discussed emerging metal nanoparticles and their effect on human health. However, it failed to present the results clearly. The gathered data in the presented figures and tables should have been discussed more. In contrast, there are too many figures and tables, it should have been narrowed down or summarized properly. In the methodology section, please cite those adapted procedures as much as possible. The same with the results and discussion section, citing discussions and comparing the obtained results with some relevant and existing studies. Some statements in the results and discussion section should have been moved or mentioned in the methodology section for simplification. My final major comment is to shorten the whole manuscript, and include only data and discussion that is meaningful in the study. Likewise, please consult a professional writer in revising the whole paper. The other comments are written below.

Specific comments:

- line 16, page 1: Spell out "ROS".

- line 97, page 3: The first figure mentioned in the article should be labeled as "Figure 1". Kindly adjust figure numbering for the rest of the manuscript.

- lines 149-150, page 4: Please clarify or expound the statement. Writing in equation form is suggested.

- lines 158-159, page 4: What do you mean by the phrase "Triplicate experiments were repeated multiple times"?

- line 173, page 5: Change the capital letter "A" to the small letter "a".

- lines 326-329, page 9: Revise the statement, not easy to understand.

- lines 337-338, page 9: Remove the statement, "We investigated the effect of each of the nanoparticles: ZnO, CuO, SnO2, and TiO2 on budding yeast at differing concentrations".

- lines 343-345, page 9: Revise the statement. What do you mean by the phrase "a cell wall which is hypothesized to be the cell structure behind no effect in viability"?

- Figure 3: Please label it properly and discussed thoroughly the meaning of each piece of data in the figures. Otherwise, could you remove those figures not discussed?

- Figure 3, lines 347-352: Most of the sentences in the figure title could be put in the discussion, revise it.

- Figure 4: What are those lines above the bar graphs? Is it p-values, how to read them?

- Figure 4, lines 363-368: Shorten the figure title, and put some of the statements in the discussion.

- lines 367-368, page 11: Remove the statement, it is already stated in the methods section.

- Figure 5: Same with the previous comment, shorten the title and move some of the statements in the discussion section. Also, remove those p-value line indicators, you could mention them during the discussion. Do these with the rest of the figures.

- Figures 6-10: Improve, check my previous comments.

- All figures and tables: Select those figures and tables that are only the highlights in the study. Remove those unwanted figures/tables. 

Author Response

Thank you very much! 

Our detailed responses are found in the attached document.

Sincerely,

Dr. Kim

Reviewer 2 Report

I am asking the authors to refer to my comments and send a reply to them.

Author Response

Thank you for the detailed comments!

All our detailed responses are included in the attached document.

Sincerely,

Dr. Kim

Round 2

Reviewer 1 Report

Major comments:

- The manuscript has improved but it needs to be trimmed down further especially on the presentation of results. Most results could be combined in a single figure or table. Please consult an expert writer regarding the coherent presentation of the manuscript.

- The figures presented in the manuscript should be reviewed, replaced, or redrawn. Most of them are not easy to read and/or understand. The title should be a simple and vivid description of the figures/images/graphs.

Other comments:

- lines 97-99, page 3: Revise the statement. Create a new "Material" subsection and list here all the purchased materials (chemicals, etc.) used in the study. 

- lines 152-154, page 4: OD corrected should be in equation format not a statement.

- line 255, page 7: Spell out "DMEM".

- Figure 1: Improve image detail of the figure, labels/units/values are hard to read.

- line 335, page 9: What do you mean by the "standard file"? Please reframe the phrase/statement.

- lines 346-347, page 9: Remove the statement in the figure title and place it in the results discussion.

- Figure 3 title: page 10: Remove the statement "ZnO, CuO, SnO2, and TiO2, and have 363 no negative impact on yeast (Saccharomyces cerevisiae) growth." and place it in the discussion. Remove the phrase "(* <0.05)" and the statement in lines 367-368.

- Figure 4 title: Move the statement in lines 383-385 in the results discussion. 

- Figure 5 title: Retain the previously removed statement in lines 406-407 to indicate the p-values.

Author Response

Please find the attached letter.

Thanks.

Kyoungtae Kim

Round 3

Reviewer 1 Report

General Comments:

- The manuscript has greatly improved. Just a few more remarks below; if these are carefully corrected, I won't have any more to contribute.

Comments:

- Abstract: Make the first statement clearer by revising it. The first two sentences might be combined.

- Lines 13-14, abstract: Up to what extent that these MNOPs do not negatively affects viability in the budding yeast, Saccharomyces cerevisiae?

Author Response

Please find the attached rebuttal letter to see the changes made in the manuscript.

Thanks,

Kyoungtae Kim
